# Fast Algorithms for $L_\infty$-Constrained S-Rectangular Robust MDPs

**Bahram Behzadian**
University of New Hampshire
bahram@cs.unh.edu

**Marek Petrik**
University of New Hampshire
mpetrik@cs.unh.edu

**Chin Pang Ho**
City University of Hong Kong
clint.ho@cityu.edu.hk

## Abstract

Robust Markov decision processes (RMDPs) are a useful building block of robust reinforcement learning algorithms but can be hard to solve. This paper proposes a fast, exact algorithm for computing the Bellman operator for S-rectangular robust Markov decision processes with $L_\infty$-constrained rectangular ambiguity sets. The algorithm combines a novel homotopy continuation method with a bisection method to solve S-rectangular ambiguity in quasi-linear time in the number of states and actions. The algorithm improves on the cubic time required by leading general linear programming methods. Our experimental results confirm the practical viability of our method and show that it outperforms a leading commercial optimization package by several orders of magnitude.

## 1 Introduction

Markov decision processes (MDPs) are a powerful framework for dynamic decision-making problems and reinforcement learning (Bertsekas and Tsitsiklis, 1996; Puterman, 2005; Sutton and Barto, 2018). The MDP model assumes that the exact transition probabilities and rewards are available. However, these transition probabilities are typically unknown and must be estimated from sampled data. Such estimations are prone to errors, and the MDP's solution is sensitive to the introduced statistical errors. In particular, the quality of the optimal policy degrades significantly even with small errors in the transition probabilities (Le Tallec, 2007).

Robust MDPs (RMDPs) mitigate MDPs' sensitivity to estimation errors by computing an optimal policy for the worst plausible realization of the transition probabilities. This set of plausible transition probabilities is known as the *ambiguity set*. In this paper, we study RMDPs with S-rectangular ambiguity sets, which can be solved in polynomial time (Hansen, Miltersen, and Zwick, 2013). However, computing the worst-case realization of transition probabilities often requires solving a linear program (LP) or another convex optimization problem. Modern solvers are powerful and efficient, but as the problem size grows, solving an LP for every state becomes computationally prohibitive (Ho, Petrik, and Wiesemann, 2018).

Most prior work has focused on RMDPs with $L_1$-constrained ambiguity sets because both convenient concentration inequalities (Weissman et al., 2003; Petrik, Ghavamzadeh, and Chow, 2016; Russel, Gu, and Petrik, 2019) and fast algorithms (Iyengar, 2005; Petrik and Subramanian, 2014; Ho, Petrik, and Wiesemann, 2020) exist for this scenario. The concentration inequalities play an important role in the data-driven construction of high-confidence RMDPs. However, ambiguity sets defined by the $L_\infty$-norm are more natural and interpretable by human modelers (Givan, Leach, and

35th Conference on Neural Information Processing Systems (NeurIPS 2021).

Dean, 2000; Delgado et al., 2016), and can significantly outperform $L_1$-based ambiguity sets in many circumstances (Behzadian et al., 2021). Unfortunately, RMDPs with S-rectangular ambiguity sets defined in terms of the $L_\infty$ ball can currently be solved only using general-purpose LP solvers, which are complex and slow.

As our main contribution, we propose a new, fast algorithm for solving RMDPs with $L_\infty$-constrained ambiguity sets. Our algorithm combines a new homotopy continuation method with a bisection method to achieve quasi-linear $\mathcal{O}(SA \log S)$ time complexity with respect to the number of states $S$ and actions $A$. This computational complexity compares favorably with the cubic $\mathcal{O}((SA)^{3.5})$ time complexity of general interior-point LP algorithms. We identify new simplifying properties of the robust optimization problem defined over $L_\infty$ balls to develop our algorithms.

Although bisection and homotopy methods have been used previously in the context of robust MDPs, their use and assumptions differ significantly from this work. A bisection method was used to solve SA-rectangular RMDPs (Nilim and El Ghaoui, 2005), but their approach does not generalize to S-rectangular RMDPs that we target. Homotopy and bisection methods have been used to solve $L_1$-constrained ambiguity sets (Ho, Petrik, and Wiesemann, 2018, 2020) but these methods are based on sparsity properties of the $L_1$-norm, which do not hold for the $L_\infty$-norm. We elaborate on this important difference after we introduce our algorithm. The existing efficient algorithms are developed for the SA-rectangular RMDPs with $L_\infty$ balls (Givan, Leach, and Dean, 2000), but they do not generalize to S-rectangular RMDPs. Developing fast optimization algorithms for S-rectangular RMDPs is especially challenging because optimal policies may need to be randomized.

Several new, fast methods have been proposed recently for solving RMDPs more efficiently. They propose replacing the standard value and policy iteration methods with more efficient algorithms, such as forms of modified policy iteration (Kaufman and Schaefer, 2013; Ho, Petrik, and Wiesemann, 2020) or gradient descent (Grand-Clément and Kroer, 2021). Most of these accelerated methods can further benefit from the fast Bellman operator algorithms that we propose in this work.

The remainder of the paper is organized as follows: Section 2 describes the basic robust MDP framework. Then, Section 3 proposes a new homotopy method for solving SA-rectangular ambiguity sets, which serves as a building block for our main contribution. In Section 4, we propose a bisection method that can solve, in combination with the homotopy method, RMDPs with S-rectangular ambiguity sets. Finally, Section 5 presents experimental results that show that our method is over 1000 times faster than using Gurobi, a leading commercial linear solver, when solving RMDPs with hundreds of states.

*Notation:* We reserve lower- and uppercase bold characters for vectors and matrices respectively. The symbol $\Delta^x$ denotes the probability simplex in $\mathbb{R}_+^x$. Finally, we use $\mathbf{I}, \mathbf{1}, \mathbf{0}$ to denote an identity matrix, a vector of ones, and a vector of zeros respectively.

## 2 Preliminaries: Robust MDPs

This section surveys the basic properties of RMDPs; for example, please see (Iyengar, 2005; Wiesemann, Kuhn, and Rustem, 2013; Ho, Petrik, and Wiesemann, 2020) for more details. We consider a finite RMDP model with states $\mathcal{S} = \{1, \ldots, S\}$ and actions $\mathcal{A} = \{1, \ldots, A\}$. The agent takes an action $a \in \mathcal{A}$ in state $s \in \mathcal{S}$, it receives a reward $r_{s,a} \in \mathbb{R}$, and it transitions to the next state $s' \in \mathcal{S}$ with a probability of $p_{s,a,s'}$. The transition probabilities $p_{s,a,s'}$ are unknown but are restricted to be in an ambiguity set $\mathcal{P} \subseteq (\Delta^S)^{S \times A}$. The initial state is distributed according to $\boldsymbol{p}_0 \in \Delta^S$.

We aim to compute a policy $\pi : \mathcal{S} \to \Delta^A$ from the set of stationary *randomized* policies $\Pi$ that maximizes the expected $\gamma$-discounted return $\rho : \Pi \times \mathcal{P} \to \mathbb{R}$ for the worst-case transition probabilities:

$$\max_{\pi \in \Pi} \min_{\boldsymbol{p} \in \mathcal{P}} \rho(\pi, \boldsymbol{p}) . \tag{1}$$

Here, $\rho(\pi, \boldsymbol{p})$ is the standard discounted infinite-horizon return for a policy $\pi$ defined as $\rho(\pi, \boldsymbol{p}) = \mathbb{E}\left[\sum_{t=0}^{\infty} \gamma^t \cdot r(S_t, A_t) \mid A_t \sim \pi(S_t), S_{t+1} \sim \boldsymbol{p}_{S_t, A_t}, S_0 \sim \boldsymbol{p}_0\right]$. The optimization problem in (1) can be seen as a zero-sum game, where adversarial nature chooses transition probabilities from the ambiguity set in order to minimize the agent's return. Since solving the general optimization problem in (1) is NP-hard (e.g., (Wiesemann, Kuhn, and Rustem, 2013)), most research has focused on RMDPs with S-rectangular and SA-rectangular ambiguity sets that can be solved in polynomial time (Iyengar, 2005; Le Tallec, 2007; Wiesemann, Kuhn, and Rustem, 2013).

*SA-rectangular ambiguity sets* $\mathcal{P}$ are defined as Cartesian products of sets $\mathcal{P}_{s,a} \subseteq \Delta^S$ for each state $s$ and action $a$ as $\mathcal{P} = \{\boldsymbol{p} \in (\Delta^S)^{S \times A} \mid \boldsymbol{p}_{s,a} \in \mathcal{P}_{s,a}, \, s \in \mathcal{S}, \, a \in \mathcal{A}\}$. The intuitive interpretation of SA-rectangularity is that nature can choose the worst transition probabilities from sets $\mathcal{P}_{s,a}$ for each state $s$ and action $a$ *independently*. We focus on ambiguity sets bounded by $L_\infty$-norm distance from nominal transition probabilities $\bar{\boldsymbol{p}}_{s,a} \in \Delta^S$ defined as

$$\mathcal{P}_{s,a} = \left\{ \boldsymbol{p}_{s,a} \in \Delta^S \mid \left\| \bar{\boldsymbol{p}}_{s,a} - \boldsymbol{p}_{s,a} \right\|_\infty \leq \kappa_{s,a} \right\}, \tag{2}$$

where $\kappa_{s,a} \geq 0$ is the robustness budget, and the nominal transition probability $\bar{\boldsymbol{p}}_{s,a}$ is typically estimated from samples of state transitions.

To streamline the definition of the robust Bellman operator, we follow the notation of Ho, Petrik, and Wiesemann (2018) and define a *nature response function* $q : \mathbb{R}_+ \times \mathbb{R}^S \to \mathbb{R}$ that represents the nature's response for a particular state $s$, action $a$, and budget $\xi$ as

$$q_{s,a}(\xi, \boldsymbol{v}) = \min_{\boldsymbol{p} \in \Delta^S} \left\{ r_{s,a} + \gamma \cdot \boldsymbol{p}^\mathsf{T} \boldsymbol{v} \mid \left\| \bar{\boldsymbol{p}}_{s,a} - \boldsymbol{p} \right\|_\infty \leq \xi \right\}. \tag{3}$$

Then, the SA-rectangular robust Bellman operator $\mathfrak{T} : \mathbb{R}^S \to \mathbb{R}^S$ for a value function $\boldsymbol{v} \in \mathbb{R}^S$ is

$$(\mathfrak{T}\boldsymbol{v})_s = \max_{a \in \mathcal{A}} \min_{\xi \leq \kappa_{s,a}} q_{s,a}(\xi, \boldsymbol{v}). \tag{4}$$

The optimal value function $\boldsymbol{v}^\star \in \mathbb{R}^S$ must satisfy the robust Bellman optimality equation $\boldsymbol{v}^\star = \mathfrak{T}\boldsymbol{v}^\star$ and can be computed using value iteration, policy iteration, or other methods (Iyengar, 2005; Ho, Petrik, and Wiesemann, 2020; Kaufman and Schaefer, 2013; Grand-Clément and Kroer, 2021).

*S-rectangular ambiguity sets* relax the assumptions of SA-rectangular sets and compute less conservative policies but with a higher computational complexity (Wiesemann, Kuhn, and Rustem, 2013). They are defined as Cartesian products of sets $\mathcal{P}_s \subseteq (\Delta^S)^A$ for each state $s$ as $\mathcal{P} = \{\boldsymbol{p} \in (\Delta^S)^{S \times A} \mid (\boldsymbol{p}_{s,a})_{a \in \mathcal{A}} \in \mathcal{P}_s, \, \forall s \in \mathcal{S}\}$. As with SA-rectangular sets, we also consider marginal ambiguity sets $\mathcal{P}_s$ defined in terms of the $L_\infty$-norm as

$$\mathcal{P}_s = \left\{ (\boldsymbol{p}_{s,a})_{a \in \mathcal{A}} \in (\Delta^S)^A \mid \sum_{a \in \mathcal{A}} \left\| \bar{\boldsymbol{p}}_{s,a} - \boldsymbol{p}_{s,a} \right\|_\infty \leq \kappa_s \right\},$$

where $\kappa_s \geq 0$ is the robustness budget and $\bar{\boldsymbol{p}}_{s,a}$ is the nominal transition probability. The important distinction from the SA-rectangular setting is that $\kappa_s$ depends only on the state and not the action. The S-rectangular Bellman operator is then defined as

$$(\mathfrak{T}\boldsymbol{v})_s = \max_{\boldsymbol{d} \in \Delta^A} \min_{\xi \leq \kappa_s} \sum_{a \in \mathcal{A}} d_a \cdot q_{s,a}(\xi, \boldsymbol{v}). \tag{5}$$

Notice that the S-rectangular Bellman operator allows for randomizing actions through the probability distribution $\boldsymbol{d}$, which improves robustness but introduces additional significant computational complexity (Wiesemann, Kuhn, and Rustem, 2013; Ho, Petrik, and Wiesemann, 2020).

The vast majority of RMDP methods employ value iteration and policy iteration principles and require computing the robust Bellman operator many times during their run (Iyengar, 2005; Wiesemann, Kuhn, and Rustem, 2013; Ho, Petrik, and Wiesemann, 2020). Therefore, it is important that it can be computed efficiently. In the remainder of the paper, we develop new quasi-linear time algorithms for computing the robust Bellman operator.

## 3 Computing the SA-Rectangular Bellman Operator in Linear Time

In this section, we develop a new quasi-linear time algorithm for computing the SA-rectangular robust Bellman operator defined by the $L_\infty$-norm. This entails solving the optimization in (4). The algorithm developed in this section also serves as the major building block of the S-rectangular algorithm described in Section 4. The remainder of the section is organized as follows: Section 3.1 first analyzes the LP formulation of the function $q$, and, then, Section 3.2 uses these properties to develop a new, fast homotopy continuation algorithm.

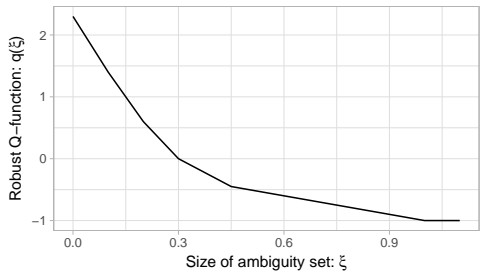
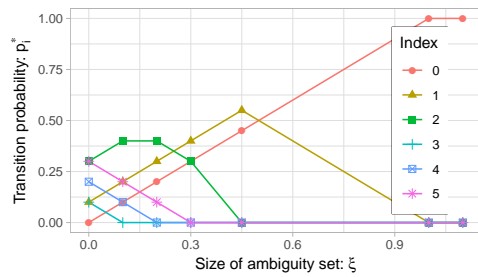

Figure 1: Function $q(\xi)$ in Example 3.1.        Figure 2: Probabilities $\boldsymbol{p}^\star(\xi)$ in Example 3.1.

Computing the SA-rectangular robust Bellman operator for a fixed state $s$, action $a$, and a value function $\boldsymbol{v}$ requires one to evaluate the nature response function $q_{s,a}(\xi, \boldsymbol{v})$ in (3). Because the symbols $s, a, \boldsymbol{v}$ are fixed throughout this section, we omit them in the notation. For example, we use $q(\xi)$ instead of $q_{s,a}(\xi, \boldsymbol{v})$ and $\bar{\boldsymbol{p}}$ in place of $\bar{\boldsymbol{p}}_{s,a}$. To further eliminate clutter, let $\boldsymbol{z} = r_{s,a} \cdot \mathbf{1} + \gamma \cdot \boldsymbol{v}$. Then, the optimization problem in (3) can be formulated as the following parametric LP:

$$
\begin{aligned}
q(\xi) &= \min_{\boldsymbol{p} \in \Delta^S} \left\{ \boldsymbol{p}^\mathsf{T} \boldsymbol{z} \mid \|\bar{\boldsymbol{p}} - \boldsymbol{p}\|_\infty \leq \xi \right\} \\
&= \min_{\boldsymbol{p} \in \mathbb{R}^S} \left\{ \boldsymbol{z}^\mathsf{T} \boldsymbol{p} \mid \mathbf{1}^\mathsf{T} \boldsymbol{p} = 1, \, -\xi \leq p_i - \bar{p}_i \leq \xi, \, p_i \geq 0, \, i = 1, \ldots, S \right\}.
\end{aligned}
\tag{6}
$$

The remainder of this section develops fast algorithms for solving (6) for all values $\xi \geq 0$.

### 3.1    Properties of Nature Response Function $q$

The LP in (6) can be solved using generic solvers, like Gurobi or Mosek, but these are impractically slow for solving RMDPs. The optimization in (6) can also be solved in quasi-linear time for any *fixed* $\xi \geq 0$, as we summarize in Appendix C. The known quasi-linear algorithm is, unfortunately, insufficient for solving the S-rectangular robust Bellman operator in Section 4. In this section, we prove results that pave the way for solving (6) for *all* $\xi \geq 0$ simultaneously in quasi-linear time, which enables efficient algorithms for both S- and SA-rectangular RMDPs.

It will be convenient to use $\boldsymbol{p}^\star(\xi)$ to refer to an optimal solution in (6). To avoid unnecessary technicalities, we assume that all elements of $\boldsymbol{z}$ are distinct, which guarantees that the optimal solution $\boldsymbol{p}^\star(\xi)$ is unique. In practice, one may add an arbitrarily small value to the elements of $\boldsymbol{z}$ to ensure that they are all distinct.

To get some intuition into the form of the nature response function $q(\xi)$ and its optimal solution $\boldsymbol{p}^\star(\xi)$, consider the following simple example.

**Example 3.1.** *Consider an* RMDP *with six states, one action,* $\boldsymbol{z} = (-1, 0, 1, 2, 3, 4)^\mathsf{T}$, *and nominal transition probabilities* $\bar{\boldsymbol{p}} = (0.0, 0.1, 0.3, 0.1, 0.2, 0.3)^\mathsf{T}$. *The functions* $q(\xi)$ *and* $\boldsymbol{p}^\star(\xi)$ *are depicted in Figures 1 and 2, where Figure 2 shows the evolution of each* $p_i(\xi)$ *using a different color for each* $i$.

The following property of the function $q$ is indispensable for our analysis and shows that $q(\xi)$ is always of the form depicted in Figure 1. It follows from standard LP properties and is proved in Appendix A.1.

**Lemma 3.2.** *The function* $q(\xi)$ *is continuous, piecewise linear, non-increasing, and convex in* $\xi$.

To develop an efficient algorithm, we now analyze the structure of the *bases* of the LP (6). Recall that a *basis* is a subset of $S$ *linearly independent* constraints in the LP, which must hold with equality. There are $S$ constraints included in each basis because $S$ is the number of optimization variables. Note that constraints may be active (or violated) without being included in the basis.

To represent a basis in (6), we use sets $\mathcal{R}_B, \mathcal{D}_B, \mathcal{N}_B, \mathcal{T}_B \subseteq \{1, \ldots, S\}$ to indicate which constraints are included in the basis with their meanings summarized in Table 1. If $i \in \mathcal{D}_B$, we call it a *donor*, if $i \in \mathcal{R}_B$, we call it a *receiver*, and if $i \in \mathcal{N}_B$, we call it a *none*. The set $\mathcal{T}_B = \{1, \ldots, S\} \setminus \mathcal{R}_B \setminus \mathcal{D}_B \setminus \mathcal{N}_B$ represents the remaining indexes, and $i \in \mathcal{T}_B$ is called a *trader*. Lemma 3.4 below justifies the names for these sets.

| Index $i \in$ | Constraints in $B$ |
|---|---|
| $\mathcal{R}_B$ (receiver) | $p_i - \bar{p}_i \leq \xi$ in $B$ |
| $\mathcal{D}_B$ (donor) | $\bar{p}_i - p_i \leq \xi$ in $B$ |
| $\mathcal{N}_B$ (none) | $p_i \geq 0$ in $B$ |

Figure 3: Composition of $B$ for $i \in \mathcal{S}$.

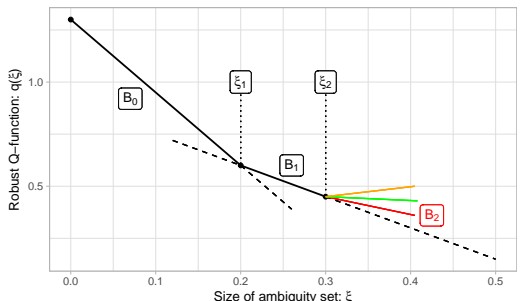

Figure 4: An illustration of Algorithm 1.

Our homotopy algorithm will leverage the specific behavior of the optimal solution $\boldsymbol{p}^\star(\xi)$ as a function of $\xi$. Because each basis $B$ represents a set of $S$ linearly independent inequalities with $S$ variables, there exists a unique solution $\boldsymbol{p}_B(\xi)$ for any value $\xi$. Note that $\boldsymbol{p}_B(\xi)$ need not be optimal or feasible.

The following lemma establishes the properties of the bases in (6) that we need to consider in our optimization; its proof can be found in Appendix A.1.

**Lemma 3.3.** *Suppose that $\boldsymbol{p}^\star$ is optimal in* (6) *for some $\xi \geq 0$. Then, there exists a basis $B$ such that* (i) $\boldsymbol{p}^\star = \boldsymbol{p}_B(\xi)$, (ii) *sets $\mathcal{R}_B, \mathcal{D}_B, \mathcal{N}_B, \mathcal{T}_B$ do not intersect,* (iii) $|\mathcal{R}_B| + |\mathcal{D}_B| + |\mathcal{N}_B| + |\mathcal{T}_B| = S$, (iv) $|\mathcal{T}_B| = 1$, *and* (v) $z_i < z_j < z_k$ *for each $i \in \mathcal{R}_B, j \in \mathcal{T}_B, k \in \mathcal{D}_B \cup \mathcal{N}_B$.*

Lemma 3.3 is important because it limits the bases relevant to the optimization, which is crucial for building fast algorithms. In particular, it shows that the sets $\mathcal{R}, \mathcal{D}, \mathcal{N}, \mathcal{T}$ *partition* the set $\mathcal{S}$, and there is always exactly one trader. The lemma also shows that $z$ coefficients for receivers are smaller than the coefficient for the trader, which is smaller than the coefficients for donors and nones.

The following lemma establishes the rate of change of the linear function $\boldsymbol{p}_B(\xi)$, which is the last necessary component for our homotopy algorithm. The lemma's proof is in Appendix A.1.

**Lemma 3.4.** *The derivatives $\dot{\boldsymbol{p}} = \nabla_\xi \boldsymbol{p}_B(\xi)$ for any basis $B$ that satisfies the properties in Lemma 3.3 are equal to*

$$\dot{p}_i = 1 \text{ if } i \in \mathcal{R}_B, \quad \dot{p}_i = -1 \text{ if } i \in \mathcal{D}_B, \quad \dot{p}_i = 0 \text{ if } i \in \mathcal{N}_B, \quad \dot{p}_i = |\mathcal{D}_B| - |\mathcal{R}_B| \text{ if } i \in \mathcal{T}_B .$$

*for $i \in \mathcal{S}$. Moreover, the slope is $\dot{q} = {}^d\!/\!d\xi\, q_B(\xi) = \sum_{i \in \mathcal{R}_B} z_i - \sum_{j \in \mathcal{D}_B} z_j + \sum_{\tau \in \mathcal{T}_B} \dot{p}_\tau z_\tau .$*

Note that Lemma 3.3 shows that each $i \in \mathcal{S}$ is either a receiver, a donor, a trader, or none. Lemma 3.4 then shows that with an increasing $\xi$, a *donor* donates its probability mass, a *receiver* receives probability mass, a *trader* either donates or receives at a variable rate, and a *none* remains unchanged.

## 3.2 Homotopy Algorithm

We are now ready to describe the proposed homotopy method and prove its correctness and complexity. Algorithm 1 summarizes a conceptual version of the homotopy algorithm. As we discuss below, one needs to avoid computing the full gradient $\nabla_\xi \boldsymbol{p}_B(\xi)$ to achieve quasi-linear time complexity. The complete algorithm with quasi-linear runtime is described in Algorithm 3 in Appendix A.2.

The main idea of Algorithm 1 is simple: it iteratively computes the linear segments of $q(\xi)$ for all $\xi \geq 0$. The algorithm starts with $\xi = 0$, where the optimal solution is $\boldsymbol{p}_0 = \bar{\boldsymbol{p}}$ with objective value $q_0 = \boldsymbol{p}_0^\top \boldsymbol{z}$. Then, the algorithm tracks the optimal bases in $q(\xi)$ as $\xi$ increases. When the $\boldsymbol{p}_{B_t}(\xi)$ becomes infeasible with the increasing $\xi$, the algorithm finds a new optimal basis $B_{t+1}$ and continues until it arrives at a basis with ${}^d\!/\!d\xi\, q(\xi') = 0$; the function $q$ is constant for all $\xi \geq \xi'$. Since $q(\xi)$ is piecewise linear in $\xi$ (see Lemma 3.2), we obtain its full description from all optimal bases.

The following theorem proves the correctness of Algorithm 1. Informally, the theorem shows that the function $q$ is piecewise linear with *breakpoints* (points of non-linearity) only at $\xi_t, t = 1, \ldots, T+1$. Note that $\xi_{T+1} = 1$ because this is the upper bound on the $L_\infty$-norm of a difference of two discrete probability distributions, and, as a result, the function $q(\xi)$ is constant for $\xi > 1$. The proof can be found in Appendix A.1.

---

**Algorithm 1** Homotopy method to compute $q(\xi)$

---

1: **input:** Objective $\boldsymbol{z}$, and nominal probabilities $\bar{\boldsymbol{p}}$ ;
2: **output:** Breakpoints $(\xi_t)_{t=0,\ldots,T+1}$ and $(q_t)_{t=0,\ldots,T+1}$ such that $q_t = q(\xi_t)$ ;
3:   Initialize $\xi_0 \leftarrow 0$, $t \leftarrow 0$, $\boldsymbol{p}_0 \leftarrow \bar{\boldsymbol{p}}$ and $q_0 \leftarrow q(\xi_0) = \boldsymbol{p}_0^{\mathsf{T}}\boldsymbol{z}$, $\tau_0 = \lceil S/2 \rceil$ and basis $B_0$ such that:
4:     $\mathcal{T}_{B_0} = \{\tau_0\}$, $\mathcal{R}_{B_0} = \{i \mid i < \tau_0\}$, $\mathcal{D}_{B_0} = \{j \mid j > \tau_0\}$, $\mathcal{N}_{B_0} = \{\}$;
5: **while** $\dot{q}_t < 0$ **do**
6:     Compute maximum step size for $B_t$ to remain feasible ($\mathcal{T}_{B_t} = \{\tau_t\}$):
7:       $\Delta\xi_t \leftarrow \max\{\xi \geq 0 \mid \boldsymbol{p}_t + \xi \cdot \nabla_\xi \boldsymbol{p}_{B_t}(\xi_t) \geq \boldsymbol{0}, \; |(\boldsymbol{p}_t + \xi \cdot \nabla_\xi \boldsymbol{p}_{B_t}(\xi_t) - \bar{\boldsymbol{p}})_{\tau_t}| \leq \xi_t + \xi\}$ ;
8:     Update breakpoints:
9:       $\boldsymbol{p}_{t+1} \leftarrow \boldsymbol{p}_t + \Delta\xi_t \cdot \nabla_\xi \boldsymbol{p}_{B_t}(\xi_t)$;     $q_{t+1} \leftarrow \boldsymbol{p}_{t+1}^{\mathsf{T}}\boldsymbol{z}$;     $\xi_{t+1} \leftarrow \xi_t + \Delta\xi_t$;
10:    Let $B_{t+1} \leftarrow$ next basis with the steepest slope (see Lemma 3.7 and Table 1);
11:    Let $t \leftarrow t + 1$ ;
12: **end while**
13: Let $\xi_{T+1} \leftarrow 1$ and $q_{T+1} \leftarrow q_T$;
14: **return:** $(\xi_t)_{t=0,\ldots,T+1}$, and $(q_t)_{t=0,\ldots,T+1}$

---

| Type | $B_{t+1}$ | $\mathcal{D}_{B_{t+1}}$ | $\mathcal{R}_{B_{t+1}}$ | $\mathcal{T}_{B_{t+1}}$ | $\mathcal{N}_{B_{t+1}}$ |
|---|---|---|---|---|---|
| 1: $\mathcal{D} \to \mathcal{N}$ | $\hat{B}^1$ | $\mathcal{D}_{B_t} \setminus \{l\}$ | $\mathcal{R}_{B_t}$ | $\mathcal{T}_{B_t}$ | $\mathcal{N}_{B_t} \cup \{l\}$ |
| 2: $\mathcal{T} \to \mathcal{N}$ | $\hat{B}^2$ | $\mathcal{D}_{B_t}$ | $\mathcal{R}_{B_t} \setminus \{m\}$ | $\{m\}$ | $\mathcal{N}_{B_t} \cup \mathcal{T}_{B_t}$ |
| 3: $\mathcal{T} \to \mathcal{D}$ | $\hat{B}^3$ | $\mathcal{D}_{B_t} \cup \mathcal{T}_{B_t}$ | $\mathcal{R}_{B_t} \setminus \{m\}$ | $\{m\}$ | $\mathcal{N}_{B_t}$ |

Table 1: Possible types of basis change at a breakpoint $\xi_{t+1}$ described in Lemma 3.7.

**Theorem 3.5.** *Suppose that Algorithm 1 returns $(\xi_t)_{t=0,\ldots,T+1}$ and $(q_t)_{t=0,\ldots,T+1}$. Then $q(\alpha \cdot \xi_t + (1 - \alpha) \cdot \xi_{t+1}) = \alpha \cdot q(\xi_t) + (1 - \alpha) \cdot q(\xi_{t+1})$ for $\alpha \in [0, 1]$ and $t = 0, \ldots, T + 1$.*

We will refer to Figure 4 in order to provide the intuition that underlies the construction of Algorithm 1 and its correctness. The figure depicts an example state of Algorithm 1 at $t = 2$ and Line 10. The solid lines show the values $q_{B_1}$ and $q_{B_2}$ when they are feasible and optimal. The dashed lines indicate when the bases are infeasible or suboptimal at each one of the breakpoints $\xi_1, \xi_2$. The colored lines at $\xi_2$ indicate the slopes for the possible candidates for $B_2$. The algorithm chooses a basis with the minimal slope.

The correctness of Algorithm 1 follows from the following three lemmas. The first lemma shows that the algorithm chooses the initial basis with the minimum possible slope.

**Lemma 3.6.** *The basis $B_0$ constructed in Line 3 of Algorithm 1 is feasible at $\xi = 0$ and has a steeper slope than any other basis $B$ that satisfies the conditions of Lemma 3.3:*

$$\frac{d}{d\xi} q_{B_0}(0) \leq \frac{d}{d\xi} q_B(0) .$$

The second lemma shows that the next basis will be selected according to one of the rules in Table 1.

**Lemma 3.7.** *Let a basis $B_t$ be optimal for $\xi_{t+1}$ in Algorithm 1, such that $\boldsymbol{p}^\star(\xi_{t+1}) = \boldsymbol{p}_{B_t}(\xi_{t+1})$ and $q(\xi_{t+1}) = q_{B_t}(\xi_{t+1})$. Assume that $\boldsymbol{p}_{B_t}(\xi)$ is infeasible for $\xi > \xi_{t+1}$. If $\mathcal{B}$ are all bases feasible for some $\xi > \xi_t$, then one with the steepest slope can be constructed as*

$$\operatorname*{argmin}_{B \in \mathcal{B}} \frac{d}{d\xi} q(\xi_{t+1}) \ni \begin{cases} \hat{B}^1 & \text{if} \quad (\boldsymbol{p}_{B_t}(\xi_{t+1}))_l = 0, \text{ for some } l \in \mathcal{D}_{B_t} \\ \hat{B}^2 & \text{if} \quad (\boldsymbol{p}_{B_t}(\xi_{t+1}))_\tau = 0, \text{ and } \mathcal{T}_{B_t} = \{\tau\} \\ \hat{B}^3 & \text{if} \quad (\bar{\boldsymbol{p}} - \boldsymbol{p}_{B_t}(\xi_{t+1}))_\tau = \xi_{t+1}, \text{ and } \mathcal{T}_{B_t} = \{\tau\} \end{cases},$$

*where $\hat{B}^1, \hat{B}^2, \hat{B}^3$ are defined in Table 1 and $m \in \operatorname{argmax}_{i \in \mathcal{R}_{B_t}} z_i$.*

Lemma 3.7 shows that there are three possible types of basis change; any other possible choice of the basis would contradict the continuity of $q(\xi)$ (Lemma 3.2). Recall also that Lemma 3.3 shows that there is always exactly one trader. The *first* type of basis change occurs when $p_l$ for a donor $l \in \mathcal{D}$ reaches zero; the donor turns into a none in the new basis. The *second* type of basis change

occurs when the trader probability mass becomes zero; the trader then turns into a none, and the receiver with the largest $z$ value becomes the new trader. The *third* type of basis change happens when the trader's gradient satisfies $d/d\xi\, p_\tau(\xi) < -1$ and its probability mass reaches to its lower bound for a given $\xi$, making the basis infeasible for greater values of $\xi$. The trader then becomes a donor, and, again, the receiver with the largest $z$ value becomes the new trader.

Finally, the third lemma shows that the optimal basis $B_t$ identified at $\xi_t$ remains feasible until $\xi_{t+1}$. Note that the convexity of $q(\xi)$ implies that the feasible basis also remains optimal.

**Lemma 3.8.** *If $B_t$ is feasible and optimal at $\xi_t$ in Algorithm 1, then it is also optimal on the interval $[\xi_t, \xi_t + \Delta\xi_t]$ computed in Line 6 of Algorithm 1.*

We now turn to the computational complexity of Algorithm 3. As the following theorem shows, the number of iterations $T$ in Algorithm 1 is at most $\mathcal{O}(S)$. Unfortunately, keeping track of $\boldsymbol{p}_t$ in each iteration of Algorithm 1 requires also $\mathcal{O}(S)$ time leading to the overall time complexity of $\mathcal{O}(S^2)$. To adapt Algorithm 1 to run in quasi-linear time, Algorithm 3, in Appendix A.2, generates the necessary values $\xi_t$, $q_t$ without tracking the complete $\boldsymbol{p}_t$ values. Its runtime is quasi-linear because it needs to sort the values of $\boldsymbol{z}$ to perform the optimization in Line 10 in constant time.

**Theorem 3.9.** *Algorithm 1 terminates in at most $\mathcal{O}(S)$ iterations and can be adapted to run in $\mathcal{O}(S \log S)$ time (see Algorithm 3 in Appendix A.2).*

We conclude by discussing the relationship with the homotopy method proposed for solving RMDPs with the $L_1$ ambiguity sets (Ho, Petrik, and Wiesemann, 2018). Although our algorithm is also a homotopy method, it is based on analysis that departs significantly from earlier work. The simplifying properties for the $L_\infty$ ambiguity sets differ significantly from the $L_1$-norm. When the ambiguity sets are defined as $L_1$ balls, only two components of $\boldsymbol{p}$ change at the time. Figure 2 illustrates that when the ambiguity sets are $L_\infty$ balls, all components of $\boldsymbol{p}$ may change with the increasing $\xi$. The fast algorithm for the $L_\infty$-constrained RMDP relies on the more subtle structure of the optimal bases described in Lemma 3.3, which leads to a more complex algorithm.

# 4 Computing the S-Rectangular Bellman Operator in Linear Time

In this section, we propose a fast algorithm for computing the robust Bellman operator (5) for S-rectangular RMDPs. We assume a fixed state $s \in \mathcal{S}$ and omit the subscripts throughout the section. For instance, the nominal probabilities for state $s$ and action $a$ are denoted by $\bar{\boldsymbol{p}}_a \in \Delta^A$. We also assume a fixed value function $\boldsymbol{v} \in \mathbb{R}^S$ and let $\boldsymbol{z}_a = r_{s,a} \cdot \mathbf{1} + \gamma \cdot \boldsymbol{v}$ for $a \in \mathcal{A}$.

The fast algorithm for computing the S-rectangular robust Bellman operator builds on Algorithm 1. As Theorem 3.9 shows, the function $q_a$ defined in (3) is piecewise linear with $\mathcal{O}(S)$ linear segments that can be computed efficiently by Algorithm 3. Since $q_a$ is piecewise linear, it is easy to construct its inverse just by swapping $\xi_t$ and $q_t$ to get the following function:

$$q_a^{-1}(u) \;=\; \min_{\boldsymbol{p} \in \Delta^S} \left\{ \|\boldsymbol{p} - \bar{\boldsymbol{p}}_a\|_\infty \;\mid\; \boldsymbol{p}^\mathsf{T} \boldsymbol{z}_a \le u \right\}, \quad \forall a \in \mathcal{A}. \tag{7}$$

The function $q_a^{-1}$ returns the budget that nature needs to achieve a response $u$. Using the function $q_a^{-1}$, we can reformulate the S-rectangular robust Bellman operator as:

$$(\mathfrak{T}\boldsymbol{v})_s = \max_{\boldsymbol{d} \in \Delta^A} \min_{\boldsymbol{\xi} \in \mathbb{R}_+^A} \left\{ \sum_{a \in \mathcal{A}} d_a \cdot q_a(\xi_a) \;\mid\; \sum_{a \in \mathcal{A}} \xi_a \le \kappa \right\} = \min_{u \in \mathbb{R}} \left\{ u \;\mid\; \sum_{a \in \mathcal{A}} q_a^{-1}(u) \le \kappa \right\}. \tag{8}$$

The correctness of this formulation follows by standard duality arguments and is proved in Lemma A.3 in Appendix A.3.

The optimization in (8) is remarkable because its objective is a one-dimensional function with one constraint. A natural algorithm to use with such an optimization problem is the bisection method outlined in Algorithm 2 (see Algorithm 4 in Appendix A.3 for a more detailed algorithm). Algorithm 2 keeps an interval $[u_{\min}, u_{\max}]$ such that the optimal $u^\star$ satisfies that $u^\star \in [u_{\min}, u_{\max}]$. In every time step, the algorithm bisects the interval $[u_{\min}, u_{\max}]$ in half and updates $u_{\min}, u_{\max}$ in order to preserve that $u^\star \in [u_{\min}, u_{\max}]$. One may think of $u_{\min}$ as the maximal known infeasible $u$ in (8) and of $u_{\max}$ as the minimal known feasible $u$ in (8).

**Algorithm 2** Bisection method for solving (7).

---

1: **input:** Desired precision $\epsilon$, functions $q_a^{-1}, \forall a \in \mathcal{A}$
2: **output:** $\hat{u}$ such that $|u^\star - \hat{u}| \le \epsilon$, where $u^\star$ is optimal in Equation (7)
3: Initialize bounds $u_{\min} \leftarrow \min_{a \in \mathcal{A}, s \in \mathcal{S}}(\boldsymbol{z}_a)_s$; $u_{\max} \leftarrow \max_{a \in \mathcal{A}, s \in \mathcal{S}}(\boldsymbol{z}_a)_s$;
4: **while** $u_{\max} - u_{\min} > 2\,\epsilon$ **do**
5:     Let $u \leftarrow (u_{\min} + u_{\max})/2$ ;
6:     **if** $\sum_{a \in \mathcal{A}} q_a^{-1}(u) \le \kappa$ **then** $u_{\max} \leftarrow u$ **else** $u_{\min} \leftarrow u$ **end if**
7: **end while**
8: **return:** $(u_{\min} + u_{\max})/2$

---

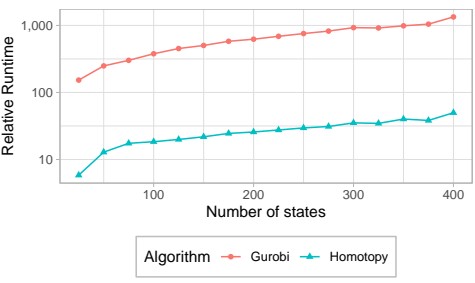 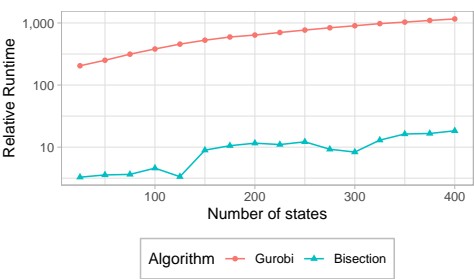

Figure 5: Relative computation time (unitless) of our algorithms and an LP solver over nominal MDP in SA-rectangular (left) and S-rectangular (right) inventory management RMDPs.

The time complexity of Algorithm 2 depends on the desired precision $\epsilon$. To remove this dependence on $\epsilon$, it is sufficient to replace the bisection by binary search over the breakpoints; we give the details of this method in Algorithm 4 in Appendix A.3. The following theorem, proved in Appendix A.3, summarizes the correctness and complexity of the proposed algorithms.

**Theorem 4.1.** *The combined Algorithms 1 and 2 compute the S-rectangular robust Bellman operator for any state $s \in \mathcal{S}$ and can be adapted (see Algorithms 3 and 4) to run in time $\mathcal{O}(SA\log(SA))$.*

## 5 Numerical Results

This section compares the empirical runtime of Algorithms 1 and 2 with the runtime of Gurobi 9.1, a leading LP solver. The results were generated on a computer with an Intel i7-9700 CPU with 32 GB RAM; the algorithms are implemented in C++.

As the main benchmark problem, we use the classic *Inventory Management (IM)* problem (Zipkin, 2000). In this problem, the decision-maker must decide at every time step how much inventory to order. The number of states and actions in this problem corresponds to the holding capacity and order size respectively. This makes it easy to scale the number of states and actions and evaluate how the algorithms scale with problem size. To evaluate the performance of our methods on small problems, we also consider the *RiverSwim (RS)* domain (Strehl and Littman, 2008) and the *Machine Replacement (MR)* domain (Delage and Ye, 2010). Please see Appendix B for the detailed description of these domains.

Figure 5 shows the time to compute the robust Bellman operator for a single state in the inventory management domain. The x-axis represents the number of states (maximum holding capacity) in the domain. The number of actions is the same as the number of states. The y-axis represents the time to compute the robust Bellman operator divided by time to compute the standard (non-robust) Bellman operator. The results show that even in MDPs with a few hundred states, the algorithms we propose are about 100 times faster than the leading LP solver. Interestingly, our algorithm is an order of magnitude faster even for small problems. We use the robustness budget $\kappa = 1.2$, but the computation time is insensitive to the particular choice of $\kappa$.

Table 2 compares the time to compute the robust policy for Machine Replacement (MR), RiverSwim (RS), and Inventory Management (IM) problems. The IM problem has 30 states. It is worth emphasizing that MR and RS are very small problems with less than 30 states, yet our algorithms are up to

| Rect. | Algorithm | MR | RS | IM |
|---|---|---|---|---|
| SA | Algorithm 1 | $< \mathbf{1}$ | **3** | **10** |
| SA | Gurobi LP | 2960 | 2240 | 9770 |
| S | Algorithm 2 | **40** | **52** | **67** |
| S | Gurobi LP | 129 | 217 | 2740 |

Table 2: Time (ms) to compute the robust policy for S- and SA-rectangular RMDPs with $L_\infty$ sets.

| Rect. | Algorithm | MR | RS | IM |
|---|---|---|---|---|
| SA | [Ho]-Alg.1 | $< 1$ | 2 | 1 |
| SA | Gurobi LP | 92 | 363 | 1140 |
| S | [Ho]-Alg.2 | 1 | 2 | 5 |
| S | Gurobi LP | 79 | 317 | 2260 |

Table 3: Time (ms) to compute the robust policy for S- and SA-rectangular RMDPs with $L_1$ sets (Ho, Petrik, and Wiesemann, 2020).

800 times faster than using an LP solver. This indicates not only that our methods scale well with the number of states but also that the constant overhead is quite small. For the sake of completeness, we include in Table 3 the timing results obtained for the RMDP with $L_1$ ambiguity sets. These results show that solving the $L_\infty$-constrained RMDP is more difficult than the $L_1$-constrained RMDP, but also that we can achieve similar dramatic speedups in $L_\infty$-constrained RMDPs as (Ho, Petrik, and Wiesemann, 2020).

## 6   Conclusion

We introduced a new homotopy method for calculating robust Bellman operators for S- and SA-rectangular ambiguity sets constructed with $L_\infty$-norm ball. Theoretically, we show that the worst-case time complexity of our algorithms is quasi-linear: $\mathcal{O}(SA\log(S))$. The algorithms also perform well in practice, outperforming a leading LP solver by several orders of magnitude.

In addition to being faster than a general-purpose LP solver, our algorithms are also much simpler. They make it possible to solve $L_\infty$-constrained RMDPs without the cost and complexity of involving a general LP solver. Although free and open-source LP solvers are available, their performance falls significantly short of commercial ones. The algorithms we propose are also easy to combine with value function approximation methods in RMDPs (Tamar, Mannor, and Xu, 2014).

In terms of future work, we believe that it is important to understand whether similar algorithms can be developed for RMDPs with more complex ambiguity sets, such as ones defined using Wasserstein distance, $L_2$-norm, or KL-divergence.

**Acknowledgments**

The authors would like to thank Bence Cserna for discussions on this topic and the reviewer for the comments that improved this paper. Partial support for the work was provided by the National Science Foundation (Grants IIS-1717368 and IIS-1815275), the CityU Start-Up Grant (Project No. 9610481), the CityU Strategic Research Grant (Project No. 7005534), the National Natural Science Foundation of China (Project No. 72032005), and Chow Sang Sang Group Research Fund sponsored by Chow Sang Sang Holdings International Limited (Project No. 9229076). Any opinion, finding, conclusion, or recommendation expressed in this material are those of the authors and do not necessarily reflect the views of the National Science Foundation and the National Natural Science Foundation of China.

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
