# A Technical Results and Proofs

## A.1 Proofs of Results in Section 3

*Proof of Lemma 3.2.* The functions $q(\xi)$ is convex due to the LP formulation of Equation (6); see (Faísca, Dua, and Pistikopoulos, 2007). $\square$

*Proof of Lemma 3.3.* (i) The statement follows from the results in sections (ii)-(v) of this lemma.

(ii) If the intersection of any pair of $\mathcal{R}_B$, $\mathcal{D}_B$, and $\mathcal{N}_B$ is not an empty set, there exist a component $i$ that satisfies two or more constraints in Table 1. In such a scenario, the basis $B$ contains linearly dependent constraints that violate the definition of a basis. $\mathcal{T}_B = \{1, \ldots, S\} \setminus \mathcal{R}_B \setminus \mathcal{D}_B \setminus \mathcal{N}_B$ by definition does not intersect with other sets.

(iii) and (iv) By definition, $B$ in $\mathcal{B}$ implies that the constraint $\mathbf{1}^\mathsf{T} \mathbf{p} = 1$ is in $B$; thus, one needs $S - 1$ additional constraints selected from Figure 3 to form a basis. However, for every $i \in [S]$, at most one of the three constraints in Figure 3 should be selected, otherwise the constraints would not be linearly independent. Therefore, it implies that there exists exactly one $j \in [S]$ such that none of the three constraints in Figure 3 is selected in $B$, and so $j \in \mathcal{T}_B$. For every $i \in [S] \setminus \{j\}$, $i \in \mathcal{R}_B \cup \mathcal{D}_B \cup \mathcal{N}_B$.

(v) We prove this results via contradiction with the following cases. Firstly, suppose there exist a basis $B'$, in which $l < \tau \in \mathcal{T}_{B'}$ where $l \in \mathcal{D}_{B'}$, then we construct another basis $B$, where $\mathcal{R}_B = \mathcal{R}_{B'} \cup \{l\}, \mathcal{D}_B = \mathcal{D}_{B'} \setminus \{l\}, \mathcal{N}_B = \mathcal{N}_{B'}$, and $\mathcal{T}_B = \mathcal{T}_{B'}$. By Lemma 3.4, we have:

$$
\begin{aligned}
\dot{q}_{B'} &= \sum_{i \in \mathcal{R}_{B'}} z_i - \sum_{j \in \mathcal{D}_{B'}} z_j + (|\mathcal{D}_{B'}| - |\mathcal{R}_{B'}|) z_\tau, \\
\dot{q}_B &= \sum_{i \in \mathcal{R}_{B'}} z_i - \sum_{j \in \mathcal{D}_{B'}} z_j + 2z_l + (|\mathcal{D}_{B'}| - |\mathcal{R}_{B'}| - 2) z_\tau
\end{aligned}
$$

and thus $\dot{q}_B - \dot{q}_{B'} = 2(z_l - z_\tau) \le 0$ as $z_l \le z_\tau$. The above construction of $B$ also ensure that $p_B(\xi)$ is feasible in a neighborhood of $\xi$, as long as $p_{B'}(\xi)$ is feasible in a neighborhood of $\xi$.

Furthermore, suppose there exist a basis $B'$, in which $l < \tau \in \mathcal{T}_{B'}$ where $l \in \mathcal{N}_{B'}$, then we construct another basis $B$, where $\mathcal{R}_B = \mathcal{R}_{B'} \cup \{l\}, \mathcal{D}_B = \mathcal{D}_{B'}, \mathcal{N}_B = \mathcal{N}_{B'} \setminus \{l\}$, and $\mathcal{T}_B = \mathcal{T}_{B'}$. By Lemma 3.4, we have:

$$
\begin{aligned}
\dot{q}_{B'} &= \sum_{i \in \mathcal{R}_{B'}} z_i - \sum_{j \in \mathcal{D}_{B'}} z_j + (|\mathcal{D}_{B'}| - |\mathcal{R}_{B'}|) z_\tau, \\
\dot{q}_B &= \sum_{i \in \mathcal{R}_{B'}} z_i - \sum_{j \in \mathcal{D}_{B'}} z_j + z_l + (|\mathcal{D}_{B'}| - |\mathcal{R}_{B'}| - 1) z_\tau
\end{aligned}
$$

and thus $\dot{q}_B - \dot{q}_{B'} = z_l - z_\tau \le 0$ as $z_l \le z_\tau$. The above construction of $B$ also ensure that $p_B(\xi)$ is feasible in a neighborhood of $\xi$, as long as $p_{B'}(\xi)$ is feasible in a neighborhood of $\xi$.

Now we prove the second part of this result.

Suppose there exist a basis $B'$, in which $m > \tau \in \mathcal{T}_{B'}$ where $m \in \mathcal{R}_{B'}$, then we construct another basis $B$, where $\mathcal{R}_B = \mathcal{R}_{B'} \setminus \{m\}, \mathcal{D}_B = \mathcal{D}_{B'} \cup \{m\}, \mathcal{N}_B = \mathcal{N}_{B'}$, and $\mathcal{T}_B = \mathcal{T}_{B'}$. By Lemma 3.4, we have:

$$
\begin{aligned}
\dot{q}_{B'} &= \sum_{i \in \mathcal{R}_{B'}} z_i - \sum_{j \in \mathcal{D}_{B'}} z_j + (|\mathcal{D}_{B'}| - |\mathcal{R}_{B'}|) z_\tau, \\
\dot{q}_B &= \sum_{i \in \mathcal{R}_{B'}} z_i - \sum_{j \in \mathcal{D}_{B'}} z_j - 2z_m + (|\mathcal{D}_{B'}| - |\mathcal{R}_{B'}| + 2) z_\tau
\end{aligned}
$$

and thus $\dot{q}_B - \dot{q}_{B'} = 2(z_\tau - z_m) \le 0$ as $z_m \ge z_\tau$. The above construction of $B$ also ensure that $p_B(\xi)$ is feasible in a neighborhood of $\xi$, as long as $p_{B'}(\xi)$ is feasible in a neighborhood of $\xi$.

$\square$

*Proof of Lemma 3.4.* Note that if $k \in \mathcal{N}_B$ implies $(\mathbf{p}_B(\xi))_k = 0$ for every $\xi$ therefore $\dot{p}_k = 0$. For all components $i \in \mathcal{R}_B$ we have $p_i - \bar{p}_i = \xi$. By taking the derivative with respect to $\xi$ we have

$\dot{p}_i = 1$. Similarly, for all $j \in \mathcal{D}_B$ we have $\bar{p}_j - p_j = \xi$. Taking the derivative leads to $\dot{p}_j = -1$. We denote by $\boldsymbol{x}_{\mathcal{G}}$ the subvector of $\boldsymbol{x} \in \mathbb{R}^S$ formed by the elements $x_i$, $i \in \mathcal{G}$, where indices are contained in the set $\mathcal{G} \subseteq \mathcal{S}$. We consider a fixed basis $B$ and thus drop the subscript $B$ for the rest of this proof.

Figure 3 implies the following useful equality that any $\boldsymbol{p}$ must satisfy.

$$1 = \mathbf{1}^\mathsf{T}\boldsymbol{p} = \mathbf{1}^\mathsf{T}\boldsymbol{p}_{\mathcal{R}} + \mathbf{1}^\mathsf{T}\boldsymbol{p}_{\mathcal{D}} + \mathbf{1}^\mathsf{T}\boldsymbol{p}_{\mathcal{N}} + \mathbf{1}^\mathsf{T}\boldsymbol{p}_{\mathcal{T}}$$
$$= \mathbf{1}^\mathsf{T}\boldsymbol{p}_{\mathcal{R}} + \mathbf{1}^\mathsf{T}\boldsymbol{p}_{\mathcal{D}} + \mathbf{1}^\mathsf{T}\boldsymbol{p}_{\mathcal{T}}$$
$$= \mathbf{1}^\mathsf{T}\boldsymbol{p}_{\mathcal{R}} + \mathbf{1}^\mathsf{T}\boldsymbol{p}_{\mathcal{D}} + p_\tau$$

where the second identity follows from the fact that $\forall k \in \mathcal{N}$ implies $p_k = 0$. By taking the derivative $\frac{d}{d\xi}$ from both sides we have:

$$0 = \mathbf{1}^\mathsf{T}\dot{\boldsymbol{p}}_{\mathcal{R}} + \mathbf{1}^\mathsf{T}\dot{\boldsymbol{p}}_{\mathcal{D}} + \dot{p}_\tau$$
$$= |\mathcal{R}| - |\mathcal{D}| + \dot{p}_\tau.$$

And finally we have:

$$\dot{q} = \boldsymbol{z}^\mathsf{T}\dot{\boldsymbol{p}}$$
$$= \boldsymbol{z}^\mathsf{T}\dot{\boldsymbol{p}}_{\mathcal{R}} + \boldsymbol{z}^\mathsf{T}\dot{\boldsymbol{p}}_{\mathcal{D}} + \boldsymbol{z}^\mathsf{T}\dot{\boldsymbol{p}}_{\mathcal{N}} + \boldsymbol{z}^\mathsf{T}\dot{\boldsymbol{p}}_{\mathcal{T}}$$
$$= \sum_{i\in\mathcal{R}} z_i - \sum_{j\in\mathcal{D}} z_j + \dot{p}_\tau z_\tau .$$

$\square$

*Proof of Theorem 3.5.* The statement is true due to linearity of $q(\xi)$ on the interval $[\xi_t, \xi_{t+1}]$ shown in Lemma 3.2, as well as the results in Lemma 3.6, Lemma 3.7, and Lemma 3.8. $\square$

*Proof of Lemma 3.6.* At $\xi = 0$, we can assume the none set is empty $\mathcal{N}_B = \emptyset$ because one can replace all non-negativity constraints $p_i \geq 0$ with $p_i - \bar{p}_i \leq \xi$ or $\bar{p}_i - p_i \leq \xi$. In Lemma 3.3, Section (v), we show for every $B \in \mathcal{B}$, $\forall i \in \mathcal{R}_B$, $\forall j \in \mathcal{D}_B$, and $\tau \in \mathcal{T}_B$ we have $i < \tau < j$. So $\dot{q}_B$ can be written as:

$$\dot{q}_B = \sum_{i\in\mathcal{R}_B} z_i - \sum_{j\in\mathcal{D}_B} z_j + (|\mathcal{D}_B| - |\mathcal{R}_B|) z_\tau$$
$$= \sum_{i=1}^{\tau-1} z_i - \sum_{j=\tau+1}^{S} z_j + ((S-\tau) - (\tau-1)) z_\tau$$
$$= \sum_{i=1}^{\tau-1} z_i - \sum_{j=\tau+1}^{S} z_j + (S - 2\tau + 1) z_\tau \quad (9)$$
$$= \sum_{k=1}^{S} \text{sign}(k-\tau) z_k + (S - 2\tau + 1) z_\tau$$

Equation (9) shows at $\xi = 0$, the trader's rate $\dot{p}_\tau = S - 2\tau + 1$. We can also show that at $\xi = 0$, for all component $i \in \{1, \dots, S\}$ we have $-1 \leq \dot{p}_i \leq 1$ because the constraints $p_i - \bar{p}_i \leq \xi$ and $\bar{p}_i - p_i \leq \xi$ are both active in equality. Thus we have

$$\min_{B\in\mathcal{B}} \quad \frac{d}{d\xi} q_B(\xi_o) = \boldsymbol{z}^\mathsf{T}\dot{\boldsymbol{p}}$$
$$\text{s. t.} \quad \mathbf{1}^\mathsf{T}\dot{\boldsymbol{p}} = 0 , \quad (10)$$
$$-\mathbf{1} \leq \dot{\boldsymbol{p}} \leq \mathbf{1} .$$

Since we previously showed the trader's exchange rate follows from $\dot{p}_\tau = |\mathcal{D}_B| - |\mathcal{R}_B|$ we can conclude $\dot{p}_\tau$ is an integer. Given the constraints in (10) at $\xi = 0$, we conclude $\dot{p}_\tau \in \{-1, 0, 1\}$. The

index of the trader is obtained from one of the following scenarios:

$$S - 2\tau + 1 = 0 \implies \tau = \frac{S+1}{2}, \tag{11}$$

$$S - 2\tau + 1 = 1 \implies \tau = \frac{S}{2}, \tag{12}$$

$$S - 2\tau + 1 = -1 \implies \tau = \frac{S+2}{2}, \tag{13}$$

When $S$ is an odd number, $\tau$ can be only $\frac{S+1}{2}$ because $S$ is also an integer and $\tau$ cannot be fractional. And when $S$ is an even number, $\tau$ can be either $\frac{S}{2}$ or $\frac{S+2}{2}$. Algorithm 3 returns the exact solution in both cases.

Given the index of trader for $B_0$, the index of all donors and receivers can be achieved form Lemma 3.2 section (v). We initialize the sets: $\mathcal{T}_{B_0} = \{\lceil S/2 \rceil\}$, $\mathcal{R}_{B_0} = \{i \,|\, i < \tau\}$, $\mathcal{D}_{B_0} = \{j \,|\, j > \tau\}$, $\mathcal{N}_{B_0} = \{\}$;

$\qquad\qquad\qquad\qquad\qquad\qquad\qquad\qquad\qquad\qquad\qquad\qquad\qquad\qquad$ $\square$

*Proof of Lemma 3.7.* Suppose $z_1 \le z_2 \le \cdots \le z_S$. Consider a base $B$ that is feasible in the neighborhood of $\xi_t > 0$, and satisfies $B = \operatorname{argmin}_{B \in \mathcal{B}} \frac{d}{d\xi} q(\xi_t)$. In Lemma 3.4, we show $\forall\, i \in \mathcal{R}_B$ and $\forall\, j \in \mathcal{D}_B \cup \mathcal{N}_B$ and $\tau \in \mathcal{T}_B$ we have $i < \tau < j$, and $\dot{q}_B$ can be written as:

$$\frac{d}{d\xi} q(\xi_t) = \dot{q}_B = \sum_{i \in \mathcal{R}_B} z_i - \sum_{k \in \mathcal{D}_B} z_k + (|\mathcal{D}_B| - |\mathcal{R}_B|)\, z_\tau \tag{14}$$

The adjacent basis $B' \in \mathcal{B}$ can be chosen from one of the following cases:

$$B' = \begin{cases} 1 & \mathcal{D}_{B'} = \mathcal{D}_B \backslash \{l\}, \quad \mathcal{N}_{B'} = \mathcal{N}_B \cup \{l\}, \quad \mathcal{T}_{B'} = \mathcal{T}_B, \quad \mathcal{R}_{B'} = \mathcal{R}_B \\ 2 & \mathcal{N}_{B'} = \mathcal{N}_B \cup \{\tau\}, \quad \mathcal{R}_{B'} = \mathcal{R}_B \backslash \{m\}, \quad \mathcal{T}_{B'} = \{m\}, \quad \mathcal{D}_{B'} = \mathcal{D}_B \\ 3 & \mathcal{D}_{B'} = \mathcal{D}_B \cup \{\tau\}, \quad \mathcal{R}_{B'} = \mathcal{R}_B \backslash \{n\} \quad \mathcal{T}_{B'} = \{n\}, \quad \mathcal{N}_{B'} = \mathcal{N}_B \\ 4 & \mathcal{R}_{B'} = \mathcal{R}_B \cup \{\tau\}, \quad \mathcal{D}_{B'} = \mathcal{D}_B \backslash \{o\}, \quad \mathcal{T}_{B'} = \{o\}, \quad \mathcal{N}_{B'} = \mathcal{N}_B \\ 5 & \mathcal{R}_{B'} = \mathcal{R}_B \cup \{\tau\}, \quad \mathcal{N}_{B'} = \mathcal{N}_B \backslash \{p\}, \quad \mathcal{T}_{B'} = \{p\}, \quad \mathcal{D}_{B'} = \mathcal{D}_B \\ 6 & \mathcal{N}_{B'} = \mathcal{N}_B \backslash \{q\}, \quad \mathcal{D}_{B'} = \mathcal{D}_B \cup \{q\}, \quad \mathcal{T}_{B'} = \mathcal{T}_B, \quad \mathcal{R}_{B'} = \mathcal{R}_B \end{cases} \tag{15}$$

Case 1 occurs when a donor becomes a none by donating all of its probability mass to a receiver. In this basis change, the index of the trader remains unchanged. $B'$ is an adjacent basis for $B$ since we only remove one active constraint ($\bar{p}_l - p_l \le \xi$), and add another one ($p_l \ge 0$). In case 2, the trader becomes a none by losing all of its probability mass. The trader's index shifts from $\tau$ to $m$, one of the receivers in $B$. Note that in case 2 also, $B'$ is an adjacent basis to $B$. We removed one active constraint ($p_m - \bar{p}_m \le \xi$), and add another one ($p_\tau \ge 0$). Case 3 is similar to case 2, however in this case the trader reaches its lower bound, and as a result the new active constraint in $B'$ is ($\bar{p}_\tau - p_\tau \le \xi$). Case 4 occurs when a trader becomes a receiver. In this scenario, the trader's index shifts from $\tau$ to $o$, which was a member of $\mathcal{D}_B$. Case 5 and case 4 are similar. However, the trader in $B'$ belongs to $\mathcal{N}_B$. In the last case, one of the components in $\mathcal{N}_B$ gain probability mass and moves to the donor's set. In the following, we show that cases 4-6 are not a feasible choice for $B'$.

Any other case violates Lemma 3.3, Section (v). The corresponding $\dot{q}_{B'}$ obtain as follows:

$$\dot{q}_{B'} = \begin{cases} 1 & \sum_{i \in \mathcal{R}_B} z_i - \sum_{k \in \mathcal{D}_B} z_k + z_l + (|\mathcal{D}_B| - |\mathcal{R}_B| - 1)\, z_\tau \\ 2 & \sum_{i \in \mathcal{R}_B} z_i - \sum_{k \in \mathcal{D}_B} z_k - z_m + (|\mathcal{D}_B| - |\mathcal{R}_B| + 1)\, z_m \\ 3 & \sum_{i \in \mathcal{R}_B} z_i - \sum_{k \in \mathcal{D}_B} z_k - z_\tau - z_n + (|\mathcal{D}_B| - |\mathcal{R}_B| + 2)\, z_n \\ 4 & \sum_{i \in \mathcal{R}_B} z_i - \sum_{k \in \mathcal{D}_B} z_k + z_\tau + z_o + (|\mathcal{D}_B| - |\mathcal{R}_B| - 2)\, z_o \\ 5 & \sum_{i \in \mathcal{R}_B} z_i - \sum_{k \in \mathcal{D}_B} z_k + z_\tau + (|\mathcal{D}_B| - |\mathcal{R}_B| - 1)\, z_p \\ 6 & \sum_{i \in \mathcal{R}_B} z_i - \sum_{k \in \mathcal{D}_B} z_k - z_q + (|\mathcal{D}_B| - |\mathcal{R}_B| + 1)\, z_\tau \end{cases} \tag{16}$$

And hence we have:

$$\dot{q}_{B'} = \begin{cases} 1 & \dot{q}_B + (z_l - z_\tau) \\ 2 & \dot{q}_B + (z_m - z_\tau)(|\mathcal{D}_B| - |\mathcal{R}_B|) \\ 3 & \dot{q}_B + (z_n - z_\tau)(|\mathcal{D}_B| - |\mathcal{R}_B| + 1) \\ 4 & \dot{q}_B + (z_o - z_\tau)(|\mathcal{D}_B| - |\mathcal{R}_B| - 1) \\ 5 & \dot{q}_B + (z_p - z_\tau)(|\mathcal{D}_B| - |\mathcal{R}_B| - 1) \\ 6 & \dot{q}_B - (z_q - z_\tau) \end{cases} \tag{17}$$

Given Lemmas A.1 and A.2, $B'_4$, $B'_5$, and $B'_6$ are not a suitable choice for $B'$ since $\dot{q}_{B'_4} \leq \dot{q}_B$, $\dot{q}_{B'_5} \leq \dot{q}_B$ and $\dot{q}_{B'_6} \leq \dot{q}_B$.

The choice over $B'_1$, $B'_2$, and $B'_3$ depend on the probability mass of the components at each break-point.

In order to minimize the decent rate in the case of $B' = B'_2$, we can show that:

$$\dot{q}_{B'} = \min_{m \in \mathcal{R}_B} \dot{q}_B + (z_m - z_\tau)(|\mathcal{D}_B| - |\mathcal{R}_B|) \tag{18}$$

We know $z_m - z_\tau \leq 0$. And $0 \leq (z_m - z_\tau)(|\mathcal{D}_B| - |\mathcal{R}_B|)$ otherwise Lemma A.2 will be violated. As a result we conclude in this particular case $(|\mathcal{D}_B| - |\mathcal{R}_B|) \leq 0$.

In order to minimize Equation (18) the term $z_m - z_\tau$ should be minimized. Since $z_1 \leq \cdots \leq z_m \leq \cdots \leq z_\tau$, therefore $m^\star = \tau - 1$. With the same reasoning we can show in the case of $B' = B'_3$ we have $n^\star = \tau - 1$.

Our results follows the continuity assumption of the solution $\boldsymbol{p}^\star = \boldsymbol{p}_B(\xi)$ for all $\xi > 0$, in which a receiver can only become a trader, not a donor nor empty, at each breakpoints. Also, a donor cannot become a receiver unless it becomes a trader first. Otherwise, the continuity assumption will be violated.

$\square$

**Lemma A.1.** *For all $B \in \mathcal{B}$ we have $|\mathcal{D}_B| - |\mathcal{R}_B| \leq 1$.*

*Proof.* Consider the problem with fixed $\xi$,

$$q(\xi) = \min_{\boldsymbol{p} \in \Delta^S} \left\{ \boldsymbol{p}^\mathsf{T} \boldsymbol{z} : \|\bar{\boldsymbol{p}} - \boldsymbol{p}\|_\infty \leq \xi \right\}, \tag{19}$$

For any fix $B \in \mathcal{B}$, we know:

$$\begin{aligned} &\text{if} \quad i \in \mathcal{R}_B \implies p_i = \bar{p}_i + \xi, \\ &\text{if} \quad j \in \mathcal{D}_B \implies p_j = \bar{p}_j - \xi, \\ &\text{if} \quad k \in \mathcal{N}_B \implies p_k = 0, \\ &\text{if} \quad \tau \in \mathcal{T}_B, \quad \exists \Delta \in \mathbb{R} \quad \text{that} \quad p_\tau = \bar{p}_\tau + \Delta. \end{aligned}$$

We also know

$$\begin{aligned} \mathbf{1}^\mathsf{T} \boldsymbol{p} = 1 \iff & \sum_{i \in \mathcal{R}_B} (\bar{p}_i + \xi) + \sum_{j \in \mathcal{D}_B} (\bar{p}_j - \xi) + \bar{p}_\tau + \Delta = 1 \\ \iff & (1 - \sum_{k \in \mathcal{N}_B} \bar{p}_k) + (|\mathcal{R}_B| - |\mathcal{D}_B|)\xi + \Delta = 1 \\ \iff & \Delta = \sum_{k \in \mathcal{N}_B} \bar{p}_k + (|\mathcal{D}_B| - |\mathcal{R}_B|)\xi \end{aligned}$$

We know for feasibility, $\Delta \leq \xi$ so we have:

$$\sum_{k \in \mathcal{N}_B} \bar{p}_k + (|\mathcal{D}_B| - |\mathcal{R}_B|)\xi \leq \xi$$

$$\sum_{k \in \mathcal{N}_B} \bar{p}_k \leq (|\mathcal{R}_B| - |\mathcal{D}_B| + 1)\xi$$

Since $\sum_{k \in \mathcal{N}_B} \bar{p}_k \geq 0$, and $\xi > 0$, we conclude $(|\mathcal{R}_B| - |\mathcal{D}_B| + 1) \geq 0$. As a result:

$$|\mathcal{D}_B| - |\mathcal{R}_B| \leq 1 .$$

$\square$

**Lemma A.2.** *let* $(\xi_t)_{t=0,\ldots,T+1}$*, and* $q(\xi)$ *is a piecewise-affine convex function with breakpoints* $\xi_l$*. Under the assumption of* $\xi_t < \xi_{t+1}$ *for all* $t = 0, \ldots, T + 1$*, we have* $\dot{q}_0 \leq \dot{q}_1 \leq \ldots \leq \dot{q}_{T+1}$*.*

*Proof.* The results follows from Theorem 24.1 in Rockafellar (1996).

$\square$

*Proof of Lemma 3.8.* The optimization problem in (3) can be formulated as the following parametric LP:

$$q(\xi) = \min_{\boldsymbol{p} \in \mathbb{R}^S} \left\{ \boldsymbol{z}^\mathsf{T} \boldsymbol{p} \mid \mathbf{1}^\mathsf{T} \boldsymbol{p} = 1, -\xi \leq p_i - \bar{p}_i \leq \xi, p_i \geq 0, i = 1, \ldots, S \right\} . \tag{20}$$

At each basis $B_t$, there are $S$ constraints that are active and satisfied in equity. In order to maintain the feasibility the basis $B_t$ on the interval $[\xi_t, \xi_t + \Delta\xi_t]$, one needs to keep track of constrains that will be violated first by increasing $\xi \in [\xi_t, \xi_t + \Delta\xi_t]$, and relax all other constraint. Since the donation rate is equal among all donors $\dot{p}_i = -1 \ \forall i \in \mathcal{D}_{B_t}$, the non-negativity constraints could be watched by following the donors with minimal probability mass $\Delta\xi_t \leftarrow \max \{\xi \geq 0 \mid \boldsymbol{p}_t + \xi \cdot \nabla_\xi \boldsymbol{p}_{B_t}(\xi_t) \geq \mathbf{0}\}$. The rate of exchange for the trader varies at each basis, as a result, the trader could violate its lower and upper bound $-\xi \leq p_\tau - \bar{p}_\tau \leq \xi$. The algorithm trace the trader's rate so one can check the constrain via $\Delta\xi_t \leftarrow \max \{\xi \geq 0 \mid |(\boldsymbol{p}_t + \xi \cdot \nabla_\xi \boldsymbol{p}_{B_t}(\xi_t) - \bar{\boldsymbol{p}})_{\tau_t}| \leq \xi_t + \xi\}$. Line 6 of Algorithm 1 combines these constraints and relaxes others.

$\square$

*Proof of Theorem 3.9.* A naive implementation of the homotopy method in Algorithm 1 has a computational complexity of $\mathcal{O}(S^2)$. The algorithm obtains the $\boldsymbol{p}^\star$ at each breakpoint. The number of iteration depends on the number of breakpoints in $q(\xi)$, which is at most $\frac{3}{2}S$. We observed numerically that the naive implementation performs on par with LP solvers and sometimes even slower. In Algorithm 3, we take advantage of the structural property of the slope of the $q$-function presented in Lemma 3.4, and only trace the optimal probability mass of the *trader* to speed up the method dramatically. Algorithm 3 compute $q$-function for each state-action pair in $\mathcal{O}(S \log S)$ for sorting the values of $\boldsymbol{z}$.

$\square$

## A.2 Detailed Homotopy Algorithm

This section provides the detailed procedure of our homotopy algorithm for computing the exact solution for robust Bellman operator with $L_\infty$ constrained ambiguity sets. Algorithm 3 starts with the initialization of the doner, receiver, and trader sets according to Lemma 3.6, and then iterates through all breakpoints. Each breakpoint has been obtained concerning the conditions that are described in Lemma 3.7. The type of each basis is change is indicated according to Table 1. We use a priority queue to keep track of the donor with the smallest probability mass. The algorithm follows the value of $q$-function at each iteration, however ignores the probability mass values for all components except the trader. The iteration stops as soon as $\xi$ exceeds the budget $\kappa$, which is given as an input.

**Algorithm 3** Homotopy method for $q(\kappa)$ with $L_\infty$ constrained ambiguity set.

---

**Input:** LP parameters $\boldsymbol{z}$, $\kappa$ and $\bar{\boldsymbol{p}}$ .
**Output:** Breakpoints $(\xi_t)_{t=0,...,T+1}$ and values $(q_t)_{t=0,...,T+1}$ ;
Initialize $\xi_0 \leftarrow 0$, $t \leftarrow 0$, $\boldsymbol{p}_0 \leftarrow \bar{\boldsymbol{p}}$ and $q_0 \leftarrow q(\xi_0) = \boldsymbol{p}_0^{\mathsf{T}}\boldsymbol{z}$ ;
Sort $\boldsymbol{z}$ in ascending order and rearrange $\bar{\boldsymbol{p}}$ accordingly
Initialize the sets: $\mathcal{T} = \{\lceil S/2 \rceil\}$, $\mathcal{R} = \{i \,|\, i < \tau\}$, $\mathcal{D} = \{j \,|\, j > \tau\}$, $\mathcal{N} = \{\}$;
$z_{\mathcal{R}} = \sum_{i \in \mathcal{R}} z_i$;    $z_{\mathcal{D}} = \sum_{j \in \mathcal{D}} z_j$
Push all elements of $\mathcal{D}$ into a min-heap $\mathcal{H}$ according to their probability mass
$\xi \leftarrow \xi_0$
**while** $\xi < \kappa$ **do**
    $\dot{p}_\tau \leftarrow |\mathcal{D}| - |\mathcal{R}|$;                                         # The trader's rate of exchange
    $j \leftarrow \mathcal{H}.top$
    $\Delta\xi_D \leftarrow p_j - \xi$
    $\Delta\xi_\tau \leftarrow$ Calculate largest feasible $\Delta p_\tau$ given $\dot{p}_\tau$
    **if** $\Delta\xi_\tau > \Delta\xi_D$ **then**
        Basis Change $\leftarrow \mathcal{D}$ *to* $\mathcal{N}$
        $\Delta\xi \leftarrow \Delta\xi_D$;
    **else**
        $\Delta\xi \leftarrow \Delta\xi_\tau$;  $p'_\tau \leftarrow p_\tau + \dot{p}_\tau \cdot \Delta\xi$;
        **if** $p'_\tau = 0$ **then**
            Basis Change $\leftarrow \mathcal{T}$ *to* $\mathcal{N}$
        **else**
            Basis Change $\leftarrow \mathcal{T}$ *to* $\mathcal{D}$
        **end if**
    **end if**
    $\Delta\xi \leftarrow \max\{\Delta\xi, \kappa - \xi\}$;
    $p_\tau \leftarrow p_\tau + \dot{p}_\tau \cdot \Delta\xi$;
    $q_t = q_{t-1} + (z_{\mathcal{R}} - z_{\mathcal{D}} + \dot{p}_\tau z_\tau) \cdot \Delta\xi$
    $\xi \leftarrow \xi + \Delta\xi$; $\xi_t \leftarrow \xi$; $t \leftarrow t+1$
    **if** Basis Change is $\mathcal{D}$ *to* $\mathcal{N}$ **then**
        $z_{\mathcal{D}} \leftarrow z_{\mathcal{D}} - z_j$;
        $\mathcal{D} = \mathcal{D}\backslash\{j\}$;
        $\mathcal{N} = \mathcal{N} \cup \{j\}$
        $\mathcal{H}.pop$
    **else**
        **if** Basis Change is $\mathcal{T}$ *to* $\mathcal{D}$ **then**
            $\mathcal{H}.push(\tau)$                               # $p = p_\tau + \xi$
            $\mathcal{D} = \mathcal{D} \cup \{\tau\}$
            $z_{\mathcal{D}} \leftarrow z_{\mathcal{D}} + z_\tau$
        **else if** Basis Change is $\mathcal{T}$ *to* $\mathcal{N}$ **then**
            $\mathcal{N} = \mathcal{N} \cup \{\tau\}$
        **end if**
        $\tau \leftarrow \tau - 1$;
        $\mathcal{T} = \{\tau\}$
        $\mathcal{R} = \mathcal{R}\backslash\{\tau\}$
        $p_\tau \leftarrow \bar{p}_\tau + \xi$
        $z_{\mathcal{R}} \leftarrow z_{\mathcal{R}} - z_\tau$
    **end if**
**end while**
The remainder of the function $q(\xi)$ will be constant: $q_{T+1} \leftarrow q_t$
$\xi_{T+1} \leftarrow \infty$
**Return:** $(\xi_t)_{t=0,...,T+1}$, and $(q_t)_{t=0,...,T+1}$

---

## A.3 Proofs of Results in Section 4

*Proof of Theorem 4.1.* The result follows from the complexity analysis of the bisection algorithm with quasi-linear time complexity in (Ho, Petrik, and Wiesemann, 2020), appendix B. □

**Lemma A.3.** *The optimal objective values of Equations* (7) *and* (8) *are equivalent.*

*Proof of Lemma A.3.* Since the functions $q_a$, for all $a \in \mathcal{A}$ in Equation (8) are convex due to the LP formulation of Equation (6). We can exchange the maximization and minimization operators in Equation (8) to obtain

$$\min_{\boldsymbol{\xi} \in \mathbb{R}_+^A} \left\{ \max_{\boldsymbol{\pi} \in \Delta^A} \left( \sum_{a \in \mathcal{A}} \pi_a \cdot q_a(\xi_a) \right) \mid \sum_{a \in \mathcal{A}} \xi_a \leq \kappa \right\}, \tag{21}$$

Since the inner maximization is linear in $\boldsymbol{\pi}$, it is optimized at an extreme point of $\Delta^A$. This allows us to re-express the optimization problem as

$$\min_{\boldsymbol{\xi} \in \mathbb{R}_+^A} \left\{ \max_{a \in \mathcal{A}} q_a(\xi_a) \mid \sum_{a \in \mathcal{A}} \xi_a \leq \kappa \right\}. \tag{22}$$

We can linearize the objective function in this problem by introducing the epigraphical variable $u \in \mathbb{R}$

$$\min_{u \in \mathbb{R}} \min_{\boldsymbol{\xi} \in \mathbb{R}_+^A} \left\{ u \mid \sum_{a \in \mathcal{A}} \xi_a \leq \kappa, u \geq \max_{a \in \mathcal{A}} [q_a(\xi_a)] \right\} \tag{23}$$

It can be readily seen that for a fixed $u$ in the outer minimization, there is an optimal $\boldsymbol{\xi}$ in the inner minimization that minimizes each $\xi_a$ a individually while satisfying $q_a(\xi_a) \leq u$ for all $a \in \mathcal{A}$. Define $g_q$ as the $a$-th component of this optimal $\boldsymbol{\xi}$:

$$g_a(u) = \min_{\xi_a \in \mathbb{R}_+^A} \{ \xi_a \mid q_a(\xi_a) \leq u \}. \tag{24}$$

We show that $g_a(u) = q_a^{-1}$. To see this, we substitude $q_a$ in Equation (24) to get:

$$g_a(u) = \min_{\xi_a \in \mathbb{R}_+^A} \min_{\boldsymbol{p}_a \in \Delta^S} \left\{ \xi_a \mid \boldsymbol{p}_a^\mathsf{T} \boldsymbol{z}_a \leq u, \|\boldsymbol{p}_a - \bar{\boldsymbol{p}}_a\|_\infty \leq \xi_a \right\}. \tag{25}$$

The identity $g_a = q_a^{-1}$ then follows by realizing that the optimal $\xi_a^\star$ in the equation above must satisfy $\xi_a^\star = \|\boldsymbol{p}_a - \bar{\boldsymbol{p}}_a\|_\infty$. Finally, substituiting the definition of $g_a$ in Equation (24) into the problem (23) show that the optimization problem (8) is equivalent to Equation (7). □

**Algorithm 4** Bisection method for the robust Bellman optimality operator (Ho, Petrik, and Wiesemann, 2020).

---

1: **Input:** Precision $\epsilon$, functions $q_a^{-1}, \forall a \in \mathcal{A}$
2: $u_{\min}$: maximum known $u$ for which Equation (7) is infeasible
3: $u_{\max}$: minimum known $u$ for which Equation (7) is feasible
4: **Output:** $\hat{u}$ such that $|u^\star - \hat{u}| \leq \epsilon$, where $u^\star$ is optimal in Equation (7)
5: **Return:** $(u_{\min} + u_{\max})/2)$
6: **while** $u_{\max} - u_{\min} > 2\,\epsilon$ **do**
7:     Split interval $[u_{\min}, u_{\max}]$ in half: $u \leftarrow (u_{\min} + u_{\max})/2$
8:     Calculate the budget required to achieve the mid-point $u$: $s \leftarrow \sum_{a \in \mathcal{A}} q_a^{-1}(u)$
9:     **if** $s \leq \kappa$ **then**
10:         $u$ is feasible: update the feasible upper bound: $u_{\max} \leftarrow u$
11:     **else**
12:         $u$ is infeasible: update the infeasible lower bound: $u_{\min} \leftarrow u$
13:     **end if**
14: **end while**

---

# B   Detailed Description of Domains

In this section, we provide a detailed description of five standard reinforcement domains that have been previously used to evaluate robustness.

As the primary metric, we compare the running time of our homotopy and bisection algorithm with Gurobi 9.1.2—a standard LP solver. In order to enable the comparison of the results among different domains, we also compare our results with the homotopy and bisection algorithm for $L_1$-constrained ambiguity sets in (Ho, Petrik, and Wiesemann, 2020).

As the first benchmark, we employ Inventory Management (IM), a classic inventory management problem (Zipkin, 2000), with discrete inventory levels $0, \ldots, S = 30$. The purchase cost, sale price, and holding cost are $2.49, 3.99$, and $0.03$, respectively. The demand is sampled from a normal distribution with a mean $S/4$ and a standard deviation of $S/6$. The initial state is 0 (empty stock). It also uses a Dirichlet prior. Table 2 summarizes the run-time for computed guaranteed returns of different methods at $0.95$ confidence levels.

The second domain is RiverSwim (RS) which is a standard benchmark (Strehl and Littman, 2008), which is an MDP consisting of six states and two actions. The process follows by sampling synthetic datasets from the true model and then computing the guaranteed robust returns for different methods. The prior is a uniform Dirichlet distribution over reachable states.

Moreover, Machine Replacement (MR) is a small benchmark MDP problem with $S = 10$ states that models progressive deterioration of a mechanical device (Delage and Mannor, 2010). Two repair actions $A = 2$ are available and restore the machine's state.

# C   Fast Algorithm for Nature Response with Fixed $\xi$

Let us consider the optimization problem (3) with fixed $\xi > 0$:

$$\min_{\boldsymbol{p} \in \Delta^S} \{\boldsymbol{p}^\mathsf{T} \boldsymbol{z} : \|\bar{\boldsymbol{p}} - \boldsymbol{p}\|_\infty \leq \xi\}, \tag{26}$$

This problem was studied by Ibaraki and Katoh (1988), and Givan, Leach, and Dean (2000). For the sake of completeness, in this section, we provide the computational procedure of solving this problem. As expressed earlier, the problem can be formulated as the following LP problem:

$$q(\xi) = \quad \min_{\boldsymbol{p} \in \mathbb{R}^S} \quad \boldsymbol{z}^\mathsf{T} \boldsymbol{p} \qquad\qquad \min_{\boldsymbol{p} \in \mathbb{R}^S} \quad \boldsymbol{z}^\mathsf{T} \boldsymbol{p}$$
$$\text{s. t.} \quad -\boldsymbol{\xi} \leq \boldsymbol{p} - \bar{\boldsymbol{p}} \leq \boldsymbol{\xi} \quad \Longleftrightarrow \quad \text{s. t.} \quad \boldsymbol{l}' \leq \boldsymbol{p} \leq \boldsymbol{u}' \tag{27}$$
$$\mathbf{1}^\mathsf{T} \boldsymbol{p} = 1, \; \boldsymbol{p} \geq \mathbf{0} \qquad\qquad \mathbf{1}^\mathsf{T} \boldsymbol{p} = 1 \; .$$

Here, $\boldsymbol{l}' = \max\{\mathbf{0}, \boldsymbol{l}\}$ and $\boldsymbol{u}' = \min\{\mathbf{1}, \boldsymbol{u}\}$ where $\boldsymbol{l} = -\boldsymbol{\xi} + \bar{\boldsymbol{p}}$ and $\boldsymbol{u} = \boldsymbol{\xi} + \bar{\boldsymbol{p}}$. The problem (27) is a bounded resource allocation problem with continuous variables, where the objective function is

convex and continuously differentiable. Without loss of generality we add the following restrictions:

First, $\boldsymbol{l}' < \boldsymbol{u}'$, since if $l_j' = u_j'$ for any $j \in \{1, \ldots, S\}$ implies that $p_j$ is fixed and can be dropped from (27). Second, $\mathbf{1}^\mathsf{T}\boldsymbol{l}' < 1 < \mathbf{1}^\mathsf{T}\boldsymbol{u}'$. Otherwise the problems is either infeasible or trivially solvable. We consider the following equivalent problem, which obtained by change in variables $\boldsymbol{x} = \boldsymbol{p} - \boldsymbol{l}'$, and the modified upper bound $\boldsymbol{u} = \boldsymbol{u}' - \boldsymbol{l}'$. Let $\alpha = 1 - \mathbf{1}^\mathsf{T}\boldsymbol{l}'$:

$$
\begin{aligned}
\min_{\boldsymbol{x} \in \mathbb{R}^S} \quad & \boldsymbol{z}^\mathsf{T}\boldsymbol{x} \\
\text{s. t.} \quad & \mathbf{0} \leq \boldsymbol{x} \leq \boldsymbol{u} \\
& \mathbf{1}^\mathsf{T}\boldsymbol{x} = \alpha .
\end{aligned}
\tag{28}
$$

To solve (28), we rely on the following relaxed problem

$$
\begin{aligned}
\min_{\boldsymbol{x} \in \mathbb{R}^S} \quad & \boldsymbol{z}^\mathsf{T}\boldsymbol{x} \\
\text{s. t.} \quad & \mathbf{0} \leq \boldsymbol{x} \\
& \mathbf{1}^\mathsf{T}\boldsymbol{x} = \alpha .
\end{aligned}
\tag{29}
$$

The above problem has a trivial solution; for example, one optimal solution is $x_i = \alpha$ for any one of $i \in \arg\min_j z_j$ and $x_j = 0$ otherwise. Therefore, one can efficiently solve the this relaxed problem (29) and check if the solution is feasible in (28). If it is feasible, then this solution is optimal in (28); otherwise, we can eliminate the associate variable $x_i$ using the following lemma.

**Lemma C.1.** *Let $\hat{\boldsymbol{x}} = (\hat{x}_1, \ldots, \hat{x}_n)$ be the optimal solution of (29). Then $\hat{x}_j \geq u_j$ implies that $x_j^\star = u_j$ holds in an optimal solution $\boldsymbol{x}^\star$ of (28).*

The proof is provided by Ibaraki and Katoh (1988). This lemma allows us to fix the optimal $x_j^\star = u_j$ and remove it from (28) and (29), which $\alpha$ should be updated and be subtracted by $u_j$. We can apply the same strategy until the optimal solution of the (29) (after removing the known optimal $x_j$'s) is also optimal in (28).