# OpenReview forum: "Fast Algorithms for $L_\infty$-constrained S-rectangular Robust MDPs"
_NeurIPS.cc/2021/Conference — NeurIPS 2021 Poster_

### Official Review · Reviewer_cHzA · 2021-07-16

**Rating:** 6
**Confidence:** 4

**Summary:**

This work studies the Robust MDPs (RMDPs) with S-rectangular ambiguity sets. The authors propose a fast algorithm to compute the robust Bellman operator over S-rectangular $L_\infty$-ball ambiguity sets, which achieves quasi-linear $\mathcal{O}(SA \log S)$ time complexity.

**Limitations And Societal Impact:**

Yes.

**Main Review:**

Originality:
In contrast with the RMDPs over $L_1$-ball ambiguity sets, the authors study the RMDPs with $L_\infty$-constrained ambiguity sets. Therefore, they propose two algorithms to solve S- and SA-rectangular RMDPs over $L_\infty$-ball ambiguity sets, which differ from the existing methods for solving RMDPs with $L_1$-constrained ambiguity sets.

Quality:
In Lines 280-281, the authors claim that "the computation time is insensitive to the particular choice of $\kappa$". However, they did not provide evidence to support the claim.

Clarity:
This paper is well written.

Significance:
This work studies the RMDPs over S-rectangular $L_\infty$-ball ambiguity sets, which are more natural and interpretable than those over $L_1$-ball ambiguity sets. Experiments demonstrate that the proposed method outperforms the leading LP solver in terms of efficiency.


**Time Spent Reviewing:**

15 hours.

---

> ### Author Response · Authors · 2021-08-09
> **Choice of $\kappa$**
>
> We would like to thank you for your time and effort in improving our paper. The computational complexity of the bisection algorithm is independent of the choice of budget $\kappa$. We also observed this behavior in our experiments.  We will elaborate on this point in the final version of this paper.

---

> ### Comment · Reviewer_cHzA · 2021-08-25
> **Response**
>
> I have read the reviewers' comments and the authors' responses. I would like to keep my score unchanged, leaning to acceptance.

---

### Official Review · Reviewer_So7E · 2021-07-16

**Rating:** 6
**Confidence:** 2

**Summary:**

This paper proposes a new algorithm that can provide fast and exact solutions for the Bellman operator for S-rectangular robust Markov decision processes with L_\infty-constrained rectangular ambiguity sets. The authors show that the solving time gain can be a cubic factor in comparing to leading general linear programming methods. This insight is further demonstrated through experimental results in comparisons to those general-purpose LP solvers.


**Limitations And Societal Impact:**

Yes

**Main Review:**

In overall, I find the paper delivered very clear motivations on why solving RMDP with L_\infty-constrained rectangular ambiguity sets is needed and challenging. It's very convincing that the proposed algorithm has come with theoretical results.

The experiment results show a lot of promising. The proposed algorithm seems to outperform the general-purpose LP solvers in terms of run time. However, given this runtime advantage, it might be also more interesting and convincing if there are results on problems with larger state spaces, at least similar to the tasks used in (Ho, Petrik, and Wiesemann, 2020) (discretized Cart-pole).




**Time Spent Reviewing:**

4

---

> ### Author Response · Authors · 2021-08-09
> **Thank you**
>
> We appreciate all valuable comments and suggestions, which helped us to improve the quality of the paper.

---

### Official Review · Reviewer_n2xU · 2021-07-19

**Rating:** 7
**Confidence:** 4

**Summary:**

The authors in the paper present a new, fast algorithm for solving RMDPs with L∞-constrained ambiguity sets. In particular, they propose a new homotopy method for solving SA-rectangular ambiguity sets and a bisection method that can solve, in combination with the homotopy method, RMDPs with S-rectangular ambiguity sets. The experimental results show that our method is over 1, 000 times faster than using Gurobi, a leading commercial linear solver when solving RMDPs with hundreds of states. Overall, this paper proposes interesting methods to faster Bellman operator algorithms, which leads to solving RMDPs more efficiently.



**Limitations And Societal Impact:**

N/A.

**Main Review:**

 Major:
*All theorems and lemmas are well-established for the new, fast algorithm for solving RMDPs with L∞-constrained ambiguity sets.
*The algorithms are well-developed and well-described.
*The structure of the method is clear-represented, and the paper is well-structured. Lemma 3.3 limits the bases relevant to the optimization, which is crucial for building fast algorithms.
*The fast algorithm for the L∞-constrained RMDP relies on the more subtle structure of the optimal bases described in Lemma 3.3, which leads to a more complex algorithm.
*The experiment proves that the algorithm is superior (simpler and faster) to a general-purpose LP solver.
*One drawback is that the paper does not discuss the non-rectangular complex ambiguity set case.

Minor:

*Line 58: …the basic Robust MDP framework.
*X label in figure 4: the size of ambiguity set -> size of ambiguity set (missing the tail of letter g while cutting the image).
*Line 225: optimizaton -> optimization.
*Line 237: a fast algorithm for compute… -> a fast algorithm for computing….

Limitations and societal impact:
The authors introduced a new, fast method for calculating robust Bellman operator for S- and SA- rectangular ambiguity sets constructed with L∞-norm ball. The algorithms are easy to combine with value function approximation methods in RMDPs. Whether similar algorithms can be developed for RMDPs with more complex ambiguity sets is a valid direction for future work.


**Time Spent Reviewing:**

10 hrs

---

> ### Author Response · Authors · 2021-08-08
> **Non-rectangular ambiguity sets**
>
> Thank you for your careful review of our paper and the comments. We have fixed the minor errors. However, we would like to clarify that the non-rectangular ambiguity sets are out of the scope of this manuscript, as the robust Bellman equation does not hold in those cases, and they are NP-hard in general.

---

> ### Comment · Reviewer_n2xU · 2021-08-11
> **Response read**
>
> I have read the authors' responses to mine and other reviewers' comments. I will maintain my score.

---

### Official Review · Reviewer_hdeg · 2021-07-19

**Rating:** 5
**Confidence:** 4

**Summary:**

In this work, authors consider S-rectangular roust MDP with L infinity constrained rectangular ambiguity and suggest a fast algorithm for computing the Bellman operator for such MDPs. Ambiguity sets are considered in the paper defined through the L-infinity norm. Specifically, in the definition, the distance between transition probability p_{s,a} and the nominal transition probability p_{s,a} is defined through L_{infinity} norm. In the prior work (Ho, Petrik and Wiesemann, 2018) instead of L_{\infinity} norm authors used L_1 norm and similarly proposed a homotopy algorithm.


**Limitations And Societal Impact:**

Limitations were not clear to me from reading the paper especially compared to previous SOTA.

**Main Review:**

I found this paper a really interesting read however, I have several concerns concerning the novelty. It is not clear to me what are the major differences to the work in (Ho, Petrik and Wiesemann, 2018). Can the authors please clarify the main contribution beyond replacing the L_1 norm from (Ho, Petrik and Wiesemann, 2018) with a L_infity norm? Does the analysis follow from (Ho, Petrik and Wiesemann, 2018) or new innovations were needed to propose the current algorithm? If so, can the authors please elaborate on what are those main differences?

1. It has not been clear to me what is the benefit (in terms of qualities of the solution when to using L_{infinity} norm instead of L_1 norm in the definition of ambiguity set? I think there has been an attempt at motivation but it was still not clear to me. Can the authors in their rebuttal clarify this through an example maybe?

2. We know that  L_{infinity} and L_1 norms are equivalent (up to a constant). Couldn't we have applied the homotopy algorithm from (Ho, Petrik and Wiesemann, 2018) to ambiguity sets defined via L_{infinity} (by appropriately scaling robustness budget \kappa_s)? If not, could the authors please clarify why not?

3) The distinction between prior work (Ho, Petrik and Wiesemann, 2018) provided in lines 228-235 was not very convincing. Especially from the point of practicality of quality of the final solution. Can the authors please help me understand this point as well?

4) On the numerical side, can we compare the authors please compared to the solution versus method in (Ho, Petrik and Wiesemann, 2018) directly in one table? Reading that part of the paper I was confused about what it is meant to say? Does this paper recover the old results, outperforms them, or is the comparison all together isn't valid to conduct? For instance, Table 2,3 does not show their relative performance, but only shows how they stand against Gurobi solver.

Minor Typos:
typo: line 24 - Algorithm 1, not Algorithm 3.
definition of \boldsymbol{z}_a uses r_{s,a}, but previously you decided to omit subscripts s.




**Time Spent Reviewing:**

3

---

> ### Author Response · Authors · 2021-08-09
> **$L_1$ vs. $L_\infty$ norm bounded ambiguity sets**
>
> We appreciate the reviewer's insightful feedback and agree that it would be helpful to demonstrate the different nature of $L_1$-norm and $L_\infty$-norm ambiguity sets; however, we referred the reader to work by Behzadian et al. (2021)* for detailed analysis on performance and problem structure of $L_1$ and $L_\infty$ sets. Under some technical conditions, robust MDPs with $L_\infty$ norm ambiguity sets are preferable with a better theoretical guarantee on performance. Some examples could also be found in Behzadian et al. (2021)*.
>
> We want to clarify that the proposed algorithm and the analysis differ significantly from the $L_1$ case. In the language of this paper, the optimal basis in the $L_\infty$ case requires multiple donors and receivers, as opposed to a single donor and receiver in the $L_1$ case; thus, both the proposed algorithm and analysis are more sophisticated than the $L_1$ case.
>
> Scaling the $L_1$ norm to approximate the $L_\infty$ case would not provide the preferred theoretical guarantee on performance (see Behzadian et al. (2021)*), so this is not preferable. It is worth mentioning that given a fixed confidence level, the  $L_\infty$ and $L_1$ ambiguity sets are not equivalent in terms of size. Thus, to achieve the same confidence level on the return, one might need to consider a relatively larger $L_1$ norm set, which decreases the performance of the robust policy.
>
> Since this paper focuses on the computation of the $L_\infty$ case, which is a different problem (in terms of computation) compared to the $L_1$ case, we did not provide the comparisons here. As mentioned above, the comparisons of the performances between the $L_1$ and $L_\infty$ case are given in Behzadian et al. (2021)*
>
> *Optimizing percentile criterion using robust MDPs. (In AISTATS-2021)

---

### Decision · Program_Chairs · 2021-09-28

**Decision:**

Accept (Poster)

**Comment:**

The reviewers and AC discussed the paper. The author response was helpful in clarifying some confusion. There was agreement that the paper is interesting. There are still some concerns about novelty and missing technical details, so this should be further clarified in the final version (in particular, the differences with Ho, Petrik and Wiesemann, 2018).

**Consistency Experiment:**

NeurIPS has a long history of experimentation. In 2014, NeurIPS ran an experiment in which 10% of submissions were reviewed by two independent committees to quantify the randomness in the review process. This year, we repeated a variant of this experiment to see how the quality of the review process has changed over time.  This paper was part of the experiment and was therefore assigned to two committees (consisting of reviewers, an Area Chair, and a Senior Area Chair) that reached independent decisions.  If both committees made the same recommendation, this recommendation was followed. If a single committee recommended acceptance, the paper was accepted (with the exception of a few cases in which the other committee identified what we considered a fatal flaw, e.g., an error in a key result).

Both committees reached the same decision: **Accept (Poster)**

The other committee assigned to the paper recommended **Accept (Poster)**.  You can find the other set of reviews, along with any follow up discussion with the authors here:
https://openreview.net/forum?id=_Eo8bl4MpT3